# Evolving cooperation in multichannel games

Kate Donahue ⬤ [1✉], Oliver P. Hauser ⬤ [2], Martin A. Nowak[3,4] & Christian Hilbe ⬤ [5✉]

Humans routinely engage in many distinct interactions in parallel. Team members collaborate on several concurrent projects, and even whole nations interact with each other across a variety of issues, including trade, climate change and security. Yet the existing theory of direct reciprocity studies isolated repeated games. Such models cannot account for strategic attempts to use the vested interests in one game as a leverage to enforce cooperation in another. Here we introduce a general framework of multichannel games. Individuals interact with each other over multiple channels; each channel is a repeated game. Strategic choices in one channel can affect decisions in another. With analytical equilibrium calculations for the donation game and evolutionary simulations for several other games we show that such linkage facilitates cooperation. Our results suggest that previous studies tend to underestimate the human potential for reciprocity. When several interactions occur in parallel, people often learn to coordinate their behavior across games to maximize cooperation in each of them.

[1] Department of Computer Science, Cornell University, Ithaca, NY 14850, USA. [2] Department of Economics, University of Exeter, Exeter EX4 4PU, UK. [3] Department of Mathematics, Harvard University, Cambridge, MA 02138, USA. [4] Department of Organismic and Evolutionary Biology, Harvard University, Cambridge, MA 02138, USA. [5] Max Planck Research Group Dynamics of Social Behavior, Max Planck Institute for Evolutionary Biology, 24306 Plön, Germany. ✉email: kdonahue@cs.cornell.com; hilbe@evolbio.mpg.de

Many of our social interactions occur in the context of repetition, which enables the evolution of cooperation by direct reciprocity[1,2]. Once there is a "shadow of the future", people are more hesitant to free ride even if there are strong short run incentives to do so. When individuals interact more than once, they can adopt conditional strategies that take into account the co-player's past behavior[3–6]. With these conditional strategies, cooperation can be enforced more effectively than would be possible in one-shot interactions. To describe direct reciprocity mathematically, researchers use the framework of iterated games[7,8]. This framework considers individuals who repeatedly engage in the same strategic interaction. Over the last decades, research on repeated games has identified which strategies can sustain cooperation[9–20], which conditions allow these strategies to spread in a population[21–27], and which of these strategies are used by human subjects[28–31].

Much of the existing literature on reciprocity is based on the assumption that individuals only engage in one repeated game. In most applications, however, people are regularly involved in multiple repeated games in parallel. Research teams routinely work on several concurrent projects[32], firms compete in distinct geographic locations[33], and political parties or entire nations need to collaborate on a whole range of different policy areas. If individuals treated all their different games as independent, each game could be analyzed in isolation, and the existing framework of direct reciprocity would continue to make correct predictions. In many scenarios, however, individuals have an incentive not to treat the different games as independent. By conditioning behavior in one game on what happened in another, individuals can increase their bargaining power[34]. This added leverage can be used to force cooperative behaviors even in those games in which cooperation is particularly difficult to sustain. To capture such strategic spillovers between distinct interactions, we introduce an evolutionary framework for multichannel games (Fig. 1).

The previous evolutionary literature has shown that remarkable dynamical effects can already occur when two or more one-shot (non-repeated) games are coupled[35–38]. This literature suggests that people find it more difficult to coordinate on an equilibrium when they interact in several games simultaneously. Evolutionary trajectories may yield persistent cycles even if each individual game has a unique absorbing state. By focusing on one-shot games, however, these previous studies do not capture reciprocal exchanges. They cannot explain how individuals optimally use one interaction to enhance cooperation in another. An independent strand of literature related to our study is previous work on multi-market price competition[39,40]. This work explores whether firms find it easier to reach collusive agreements when they are in contact in several distinct markets. The corresponding models suggest that multi-market contact may help, but only if there is sufficient heterogeneity between firms or markets[39], or if monitoring is imperfect[40]. Importantly, however, these models take a static approach. By constructing specific collusion strategies, they identify conditions under which multi-market contact alters the possible equilibrium outcomes (see Supplementary Note 1 for a more detailed description). In contrast, we take an evolutionary approach. We are interested in the strategies that the players themselves adopt over time, when given the choice between different strategies of similar complexity.

Our evolutionary findings suggest that individuals quickly learn to coordinate their own behaviors across different social dilemmas. They tend to use cooperation in more valuable interactions as a means to promote cooperation in those games with a larger temptation to defect. Remarkably, this endogenous coupling of independent games does not need to come at the cost of reduced cooperation in the most valuable game. Instead, individuals often evolve to be more cooperative in all games, including

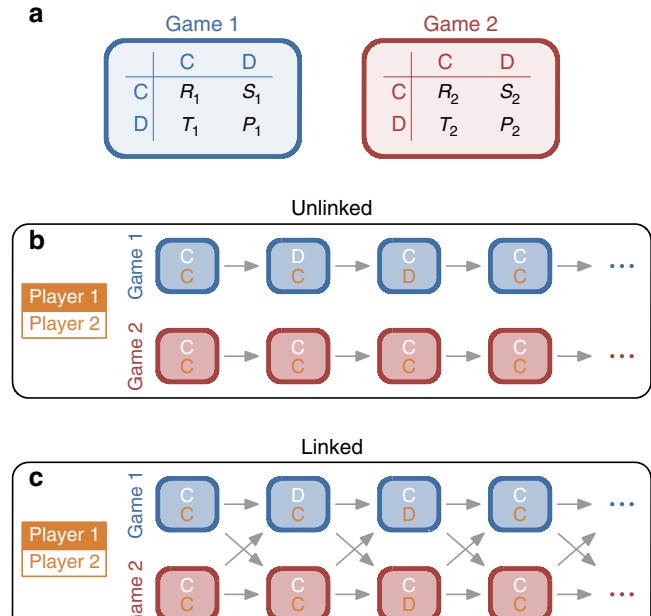

**Fig. 1 Cooperation in multichannel games. a** In a multichannel game, individuals repeatedly interact in several independent games. Here, we illustrate the case of two players who interact in two different prisoner's dilemma games. In each game, players can either cooperate (C) or defect (D). A player's payoff in each game $k$ is either $R_k$, $S_k$, $T_k$, or $P_k$, depending on the player's and the co-player's decision. In the main text, we consider the case that the two games take the form of a donation game[7], such that $R_k = b_k - c_k$, $S_k = -c$, $T_k = b_k$, and $P_k = 0$, where $b_k$ and $c_k$ are the benefit and cost of cooperation. The effect of other payoffs is studied in the Supplementary Information. Players interact for infinitely many rounds. In each round, players simultaneously determine how to act in each game. We distinguish two different scenarios. **b** In the unlinked case, the two players are restricted to treat each game as independent. They only react to the co-player's previous action in the very same game. **c** In the linked case, the two players are able to couple the two games—they are allowed to react to a co-player's defection in one game by defecting in the other. In the above example, the first player defects in the first game in the second round (as indicated by the white D in the respective blue box). In response, the second player defects in both games in the third round.

those in which subjects are highly cooperative even without any linkage. To explore this effect in more detail, we explore which strategies can be used to sustain full cooperation in all concurrent games. When each game is a donation game, we provide a complete characterization of these "partner" strategies. Based on these analytical results, we show that the set of partner strategies expands considerably once individuals are allowed to link their different games. Our findings suggest that linkage enhances the influence and flexibility individuals have. This enhanced flexibility is crucial to establish cooperation in some games, and it can further promote the already existing cooperative behaviors in others.

## Results

**An evolutionary framework for multichannel games.** In the main text, we introduce our framework for the most simple setting, by considering two players who simultaneously interact in two games, as depicted in Fig. 1a (further generalizations are discussed in the Supplementary Note 2). For each game, players independently decide whether they cooperate (C) or defect (D). Payoffs take the form of a so-called donation game[7]. That is, a player who cooperates in game $k$ transfers a benefit $b_k$ to the

co-player at own cost $c_k$. Defectors pay no cost and create no benefit. We assume $b_k > c_k > 0$, such that each game has the incentive structure of a prisoner's dilemma. It follows that cooperation does not evolve in either game if players only interact once[35]. However, here we consider repeated interactions. After each round, there is another one in which players again have to decide whether to cooperate in either game. We refer to such repeated interactions across multiple parallel games as a multi-channel game. A player's payoff in the multichannel game is computed by summing up her average payoffs across all individual games.

To explore the effect of endogenous linkage, we distinguish between two versions of multichannel games. In the unlinked case (Fig. 1b), individuals consider each game in isolation. To decide whether to cooperate in game $k$, they only take into account what previously happened in that very game, while ignoring what happened in the other. Such a scenario may reflect, for example, two companies who compete in two different geographic markets, each managed by independent subunits. In contrast, in the linked case (Fig. 1c), players are able to react in each game to what previously happened in all games. In particular, players have multiple opportunities to retaliate against a co-player who defected in one of the games. They may either respond by defecting in the same game, in the other, or in both.

Because players may condition their behavior on the entire previous history of play, strategies for multichannel games can be arbitrarily complex. To make a computational analysis of the evolutionary dynamics feasible, we assume that players choose from a predetermined set of given complexity. Here, we first consider reactive strategies[7]. A player's behavior in any given round may thus depend on the co-player's action in the last round, but it is independent of all previous rounds. In the unlinked case, reactive strategies can be represented by 4-tuples

$$\mathbf{p} = (p_C^1, p_D^1; p_C^2, p_D^2). \quad (1)$$

Here, $p_a^k$ is a player's probability to cooperate in game $k$, dependent on the co-player's previous action $a \in \{C, D\}$ in that game. In the linked case, reactive strategies take the form

$$\mathbf{p} = (p_{CC}^1, p_{CD}^1, p_{DC}^1, p_{DD}^1; p_{CC}^2, p_{CD}^2, p_{DC}^2, p_{DD}^2). \quad (2)$$

Now, $p_{a_1 a_2}^k$ is the probability to cooperate in game $k$ depending on the co-player's previous actions in game one and two, respectively. In the linked case, players themselves may decide to treat each game as independent, by choosing a strategy for which

$$p_{CC}^1 = p_{CD}^1, \quad p_{DC}^1 = p_{DD}^1, \quad p_{CC}^2 = p_{DC}^2, \quad p_{CD}^2 = p_{DD}^2. \quad (3)$$

It follows that the set of linked strategies (2) contains the unlinked strategies (1) as a (strict) subset. In the following, we explore the effect of linkage in two ways. (i) We compare the evolving cooperation rates between the linked and unlinked case; and (ii) we analyze to which extent players in the linked case use strategies that are infeasible in the unlinked case.

To describe how players adapt their strategies over time, we consider a pairwise comparison process[41,42]. Evolution occurs in a population of fixed size $N$. Players receive payoffs by interacting with all other population members. Occasionally, they are given a chance to update their strategies. With probability $\mu$ (reflecting a mutation probability), players do so by random strategy exploration. In that case, they choose a new strategy uniformly at random from the set of all available strategies. Otherwise, with probability $1 - \mu$, players consider imitating the strategy of someone else. To this end, they randomly sample a role model from the population. Then they adopt the role model's strategy with a probability that increases in the role model's payoff (for

details, see "Methods"). Over time, the two elements of imitation and random strategy exploration yield a stochastic process on the space of all possible population compositions. We explore this process through computer simulations in the limit of rare mutations[43–46] (the respective code is provided in Supplementary Note 5).

**The effect of linkage in concurrent prisoner's dilemma games.** To explore evolution in multichannel games, we have first run simulations for a scenario in which the first game has a higher benefit of cooperation, such that $b^1 > b^2$. When the two games are unlinked (Fig. 2a), individuals quickly tend to cooperate in the first game (74.1%) but less so in the second (37.5%). Instead, when the two games are linked (Fig. 2b), cooperation in the second game increases considerably (to 64.4%), but also the cooperation rates in the first game show a moderate increase (to 87.2%). To explore these effects in more detail, we have recorded which behaviors the players exhibit by the end of each simulation. We distinguish four classes, depending on whether individuals tend to cooperate in both games, cooperate in one game but defect in the other, or defect in both (Fig. 2c, d). In the unlinked case, the most abundant behavior is to cooperate in the more valuable game and to defect in the other. Only if the two games are linked, most players coordinate on mutual cooperation in both games.

To understand how linkage facilitates the evolution of mutual cooperation, we have recorded which strategies the players use. In the unlinked case, cooperating players use strategies resembling Generous Tit-for-Tat[3,4] (Fig. 2e). They fully reciprocate a co-player's cooperation in the respective game, but they still cooperate with some positive probability if the co-player defects. In the linked case, the evolving strategies are similar, with one crucial exception. If the co-player cooperated in one game but not in the other, individuals react with a reduced cooperation probability in both games, independent of where the transgression occurred (Fig. 2f). We refer to such strategies as Linked Tit-for-Tat (LTFT). Individuals who adopt LTFT have learned to connect the two games. Their actions in either game depend on what happened in the other.

**Characterization of partners, semi-partners, and defectors.** To explore the emergence of linkage in more detail, we have mathematically characterized the strategy classes that give rise to the four possible behaviors described above. We say a strategy is a partner if two individuals with that strategy cooperate in both games and if the respective strategy is a Nash equilibrium (such that no player has an incentive to deviate). Similarly, we say a strategy is a game-$k$ semi partner if it gives rise to a Nash equilibrium where the two players cooperate in game $k$ but defect in the other. Finally, a strategy is a defector if it gives rise to a Nash equilibrium with mutual defection in both games. For repeated games, the respective strategy classes of partners and defectors have been characterized recently[13–17]. Here we describe them for multichannel games. We recover the previous work as a special case (see Supplementary Note 3 for details). In the unlinked case, we find that a reactive strategy is a partner only if for both games $k$,

$$p_C^k = 1$$
$$p_D^k \leq 1 - \frac{c_k}{b_k}. \quad (4)$$

Supplementary Fig. 1 gives a graphical illustration. The first condition ensures that players are mutually cooperative, while the second condition guarantees that no other strategy can invade (not even strategies of higher complexity). In the linked case, we

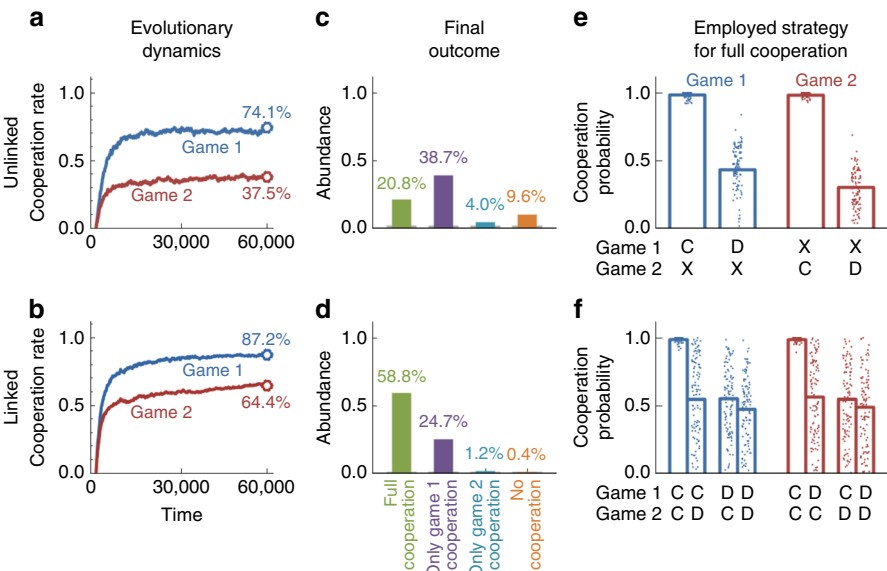

**Fig. 2 The evolutionary advantage of linkage. a**, **b** We simulated the dynamics when players simultaneously engage in a game with a high benefit of cooperation (Game 1, blue) and a game with a comparably low benefit (Game 2, red). We find that linking has a strongly positive effect in the low-benefit game and a weakly positive effect in the high-benefit game. **c**, **d** We recorded which behaviors the players exhibit by the end of each simulation. To this end, we define a strategy to be cooperative in a given game if the respective cooperation rate against itself is at least 80%. Similarly, we say a strategy is non-cooperative, if this cooperation rate is below 20%. This distinction gives rise to four behavioral classes, depending on whether players are cooperative in both games, cooperative in one game but non-cooperative in the other, or non-cooperative in both. Only when the two games are linked, players are most likely to be fully cooperative in both. **e**, **f** We analyze which strategies the players use when they are fully cooperative. Each bar shows the respective mean value, whereas dots represent 100 randomly sampled realizations of the simulation. In the linked case, players only exhibit a high mean cooperation probability (in either game) if their co-player previously cooperated in both games. As parameters we used $b_1 = 5$, $b_2 = 3$, and $c_1 = c_2 = 1$, in a population of size $N = 50$ using a selection strength parameter $s = 2$. The figure shows averages over 1000 simulations of the pairwise comparison process[41,42] in the limit of rare mutations (see "Methods" for details).

find that a partner strategy needs to satisfy

$$p^1_{CC} = p^2_{CC} = 1$$

$$\frac{b_1}{b_1 + b_2} \cdot p^1_{DC} + \frac{b_2}{b_1 + b_2} \cdot p^2_{DC} \leq 1 - \frac{c_1}{b_1 + b_2}$$

$$\frac{b_1}{b_1 + b_2} \cdot p^1_{CD} + \frac{b_2}{b_1 + b_2} \cdot p^2_{CD} \leq 1 - \frac{c_2}{b_1 + b_2} \qquad (5)$$

$$\frac{b_1}{b_1 + b_2} \cdot p^1_{DD} + \frac{b_2}{b_1 + b_2} \cdot p^2_{DD} \leq 1 - \frac{c_1 + c_2}{b_1 + b_2}$$

These conditions are visualized in Supplementary Fig. 2. By comparing (4) with (5), we can explore why linkage facilitates the evolution of mutual cooperation. In the unlinked case, every single cooperation probability $p^k_D$ needs to fall below a certain threshold. In particular, in neither game are the players allowed to be more generous than prescribed by the conventional Generous Tit-for-Tat strategy[3,4]. In contrast, in the linked case the respective thresholds only need to be met on average, when taking a weighted mean across both games. Players can afford to be more forgiving in one game by being more restrictive in the other. The specific weights depend on how valuable cooperation is in the respective game. The more valuable, the less forgiving a player should be after a co-player's defection.

Conditions (4) and (5) can also be used to calculate how likely it is that a randomly chosen cooperative strategy is a partner (see Supplementary Note 3 for details). This calculation confirms that random strategy exploration is more likely to generate partner strategies when the two games are linked (Supplementary Fig. 3a). Linkage is particularly advantageous when the two games differ in their benefit (Supplementary Fig. 3b). In that case, partner strategies are rare in the unlinked case where cooperation in the low-benefit game is difficult to sustain. In the linked case, on the other hand, players only need to slightly adapt their cooperation probabilities in the high-benefit game to also sustain cooperation

in the other. For semi-partners and defectors, linkage has the opposite effect. These strategies tend to become less abundant when the games are linked (Supplementary Fig. 3c–h and Supplementary Note 3 for details).

**Partners, semi-partners, and defectors in evolution.** In a next step, we have investigated to which extent the four strategy classes described above can explain the simulation results in Fig. 2. To this end, we have run further simulations in which we record how often evolving populations learn to adopt strategies in the neighborhood of each strategy class (for details, see Supplementary Note 3). In the absence of any selection pressure, the four classes only amount to a negligible fraction of all observed behaviors (Fig. 3a, d). But once evolution is determined by a strategy's relative success, the four strategy classes account for 72% of the observed behaviors in the unlinked case (Fig. 3b) and for more than 95% in the linked case (Fig. 3e). In the unlinked case, we mainly observe three behaviors: partners who cooperate in both games, semi-partners who only cooperate in the more profitable first game, and other (unclassified) strategies. In the linked case, partners predominate.

This analysis also shows how resistant strategies from different strategy classes are to mutant invasions (Fig. 3c, f). In the unlinked case, it takes ~1000 attempts by randomly generated mutant strategies to invade a resident partner or game-1 semi-partner strategy. Once the two games are linked, all named strategy classes become more resistant, but partners particularly so. Now, it takes on average more than 18,000 mutants until a resident partner is invaded, and successful mutants are more likely to be partners again. To further corroborate these findings, we have systematically varied the benefit of cooperation in the first game (Supplementary Figs. 4 and 5). Throughout we find

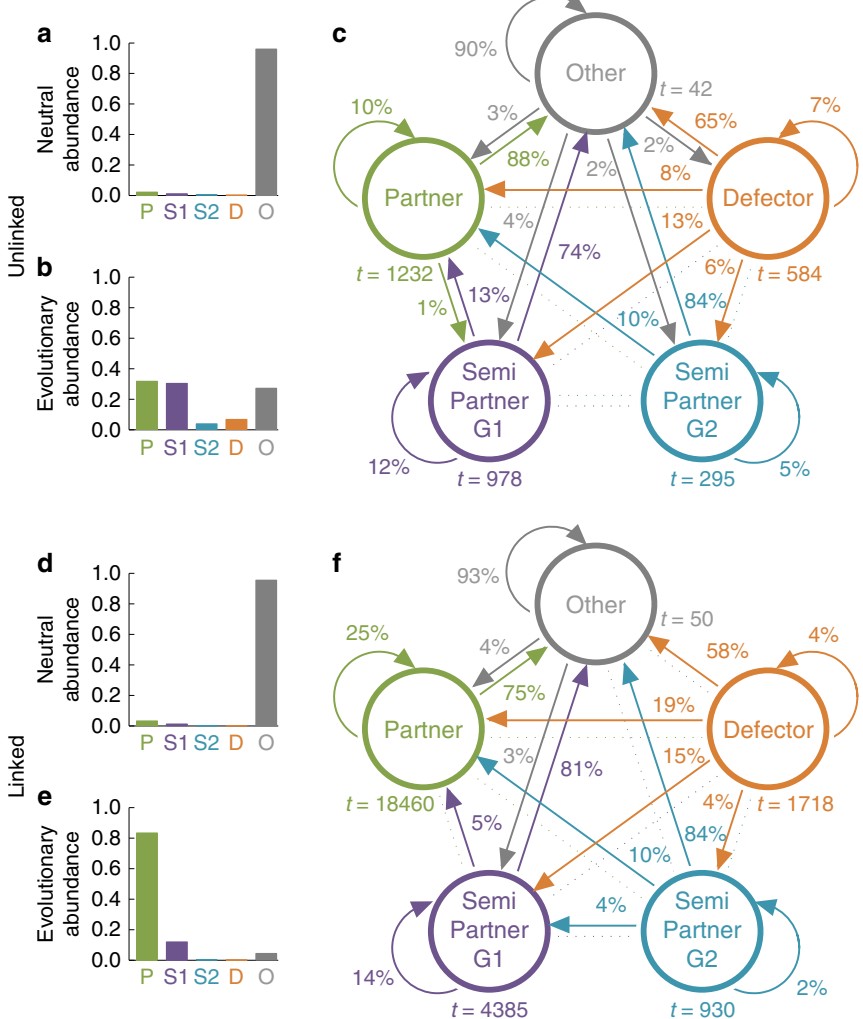

**Fig. 3 Linked games favor partner strategies.** To understand how linking promotes cooperation, we distinguish between different strategy classes, representing different kinds of Nash equilibria. Partners mutually cooperate in both games; Game-$k$ semi-partners cooperate in game $k$ but defect in the other; and Defectors defect in both games. Each strategy class can be characterized analytically. In addition, we consider the class of "others", which includes all remaining strategies. **a**, **d** Partners, semi-partners, and defectors only make up a minor fraction of the entire strategy space. **b**, **e** However, when strategies result from an evolutionary process that favors strategies with high payoffs, these strategies account for >72% (unlinked) and for >95% (linked) of the observed behaviors. **c**, **f** For each resident strategy that emerged during the simulation, we classified its type, recorded the time $t$ (number of mutants) until invasion, and classified the successful mutant. In the unlinked case, partners and game-1 semi-partners are most robust to invasion. In addition, other strategies are frequently played despite their poor robustness. In the linked case, partners are clearly favored. The figure is based on the same parameters as in Fig. 2, but simulations are run for longer (during each simulation, we consecutively introduce $2 \times 10^7$ mutants; the figure shows data from eight independent simulations). Because the first four strategy classes have measure zero, we have recorded how often players are in a small neighborhood of the respective strategy class (see Supplementary Note 3).

that linkage leads to more cooperation in both games, driven by a higher abundance of partner strategies. Our results are independent of the considered evolutionary parameters, such as population size, selection strength, frequency of mutations, or error rate (Supplementary Fig. 6).

**Evolution among memory-1 players.** After exploring the effects of linkage among reactive players, we have run simulations for memory-1 strategies (Fig. 4 and Supplementary Fig. 7). In addition to the co-player's actions in the last round, a memory-1 player also takes her own actions into account. Previous research suggests that with memory-1 strategies, players should learn to adopt Win-Stay Lose-Shift (WSLS). In each individual game they should repeat their previous action if it yielded a positive payoff, and they should switch to the opposite action otherwise. While our simulations generate WSLS strategies in the unlinked case,

players in the linked case rather adopt a rule we term Cooperate if Coordinated (CIC). A player with this strategy cooperates in all games if the players' previous actions in each game coincided. In Supplementary Note 4, we prove that CIC can establish full cooperation under conditions where WSLS fails. Moreover, we show that CIC is most valuable when there is considerable heterogeneity among the games individuals play.

## Discussion
Herein, we have introduced a general framework to explore the evolution of reciprocity when people interact in multiple games simultaneously. Such multichannel games are different from usual repeated games because they allow players to engage in cross-reciprocity. If a player defects in one game, the co-player may respond by defecting in the same game, a different game, or in all games currently played. Previous work suggests that these

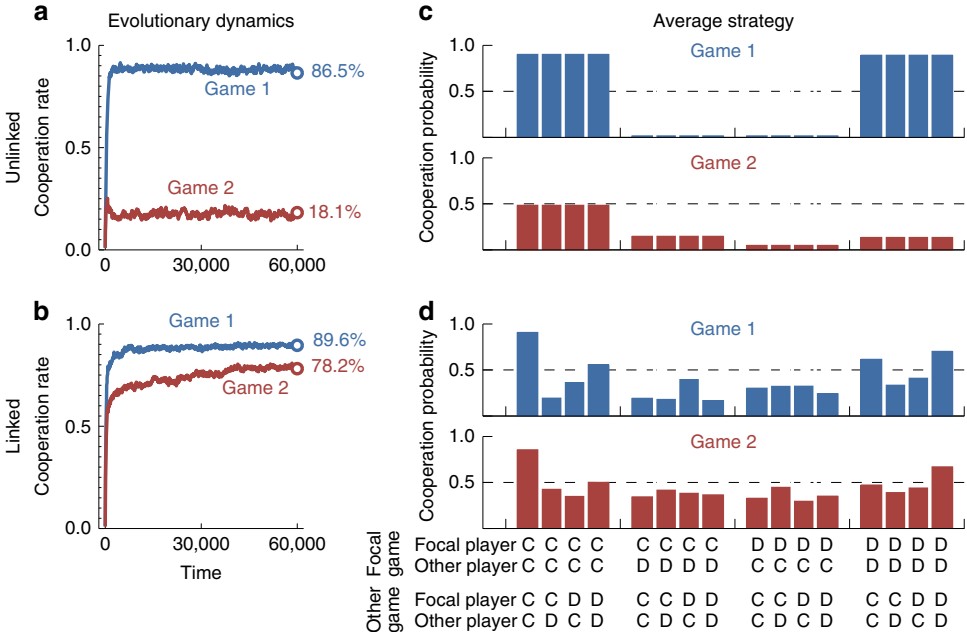

**Fig. 4 Multichannel games among memory-1 players. a, b** The positive effects of linkage are even stronger when players are able to choose among all memory-1 strategies. **c** In the unlinked case, players tend to use different strategies for the two games for the respective parameter values. In the first game, their strategy resembles Win-Stay Lose-Shift[5]. They are most likely to cooperate if either both players cooperated in the previous round, or both players defected. In the second game, the average strategy resembles Grim[7]. Here, a player cooperates only if both players did so in the previous round. **d** In contrast, in the linked case, players use similar strategies for both games. Players are most likely to cooperate if in each previous game they used the same action (either CC or DD; the action played may be different in different games). We call the respective strategy Cooperate if Coordinated (CIC). As parameters we used $b_1 = 4$, $b_2 = 2$, and $c_1 = c_2 = 1$, $N = 50$, and $s = 1$. Players can choose among all deterministic memory-1 strategies, for which the conditional cooperation probability is either zero or one. The strategies are subject to errors, using an error rate $\varepsilon = 0.01$. The figure shows averages over 500 independent simulations.

added retaliation opportunities do not necessarily enhance cooperation[39]. After all, when multiple social dilemmas occur in parallel, this does not only increase the opportunities to retaliate, but also the opportunities to defect in the first place. As a result, merely interacting across several copies of the same social dilemma does not alter the possible equilibrium outcomes[39].

Using an evolutionary approach, we nevertheless find that multichannel games facilitate cooperation. Even in those cases in which linkage leaves the set of equilibrium outcomes unchanged, it may still affect the number of strategies that give rise to each equilibrium outcome. To illustrate this point, we have considered a multichannel game in which individuals simultaneously interact in multiple social dilemmas. Once players are able to link these games, the set of partner strategies that enforce full cooperation increases substantially (even if all games coincide). As a consequence, players are more likely to discover and adopt these partner strategies over the course of evolution.

Throughout the main text, we have focused on simple instances of multichannel games. We have considered two players who use reactive or memory-1 strategies to interact across two donation games. However, our framework is in no way restricted to these cases. In the "Methods", we describe how our model can be adapted to cover interactions across arbitrarily many donation games. The respective characterizations of partners, semi-partners and defector strategies immediately carry over, and also the evolutionary dynamics is similar (see Supplementary Fig. 8 and Supplementary Note 3).

In addition, we have also explored the dynamics of social dilemmas in which mutual cooperation is no longer the uniquely optimal outcome[47], including the snowdrift game[48,49] and the volunteer's dilemma[50] (Supplementary Figs. 9 and 10). Again, we observe higher payoffs when players are able to link two

independent instances of these game classes. However, the players' payoffs may no longer approach the social optimum even for substantial benefits of cooperation. Finally, we have also explored cases in which at least one of the concurrently played games takes the form of a coordination game. Coordination games allow for full cooperation even without linkage, and in fact without any repeated interactions. Taking the so-called sculling game[51] as an example, we show that in such cases, linkage can be detrimental. Especially if cooperation is risk-dominant, high cooperation rates can already be achieved in the unlinked case. Here, linking leads to a slight decrease in cooperation rates (Supplementary Figs. 9 and 10). We conclude that strategic linkage is most effective in strict social dilemmas, in which repeated interactions are key to sustain cooperation.

Although groups of individuals often engage in several interactions in parallel, traditional models tend to explore each of these interactions as an isolated game. Our work suggests that such models may underestimate the human potential for cooperation. Once individuals are allowed to link their concurrently ongoing interactions, they often learn to coordinate their behavior across games in order to enhance cooperation in each of them.

## Methods

**Multichannel games**. We provide a full account of the applied methods and the proofs of our mathematical results in the Supplementary Information. Here we provide a summary of the considered setup and the respective findings.

In a multichannel game, a group of individuals repeatedly interacts in several independent (elementary) games, as depicted in Fig. 1. Here, we discuss the special case that the group consists of two individuals who interact in $m$ games, where each game takes the form of a social dilemma. In the main text we describe our results for $m = 2$ games. Generalizations are presented in the Supplementary Information.

In each round, players decide whether to cooperate (C) or to defect (D) for each of the $m$ games. Games are independent in the sense that a player's one-round

payoff in each game only depends on the player's and the co-player's action in that game, irrespective of the outcome of the other games. For each game $k$, we denote the possible one-round payoffs by $R_k$, $S_k$, $T_k$, and $P_k$. Here, $R_k$ is the reward when both players cooperate, $S_k$ is the sucker's payoff a cooperator obtains when the co-player defects, $T_k$ is the temptation to defect when the co-player cooperates, and $P_k$ is the punishment payoff for mutual defection. For the game to be a social dilemma[52,53], we assume that $R_k > P_k$ (such that mutual cooperation is favored to mutual defection), and that either $T_k > R_k$ or $P_k > S_k$. The prisoner's dilemma corresponds to the case where all three inequalities are satisfied. Throughout the main text, we focus on a special case of the prisoner's dilemma, called donation game[7]. In the donation game, cooperation means to pay a cost $c_k > 0$ to transfer a benefit $b_k > c_k$ to the co-player. It follows that $R_k = b_k - c_k$, $S_k = -c_k$, $T_k = b_k$, and $P_k = 0$. However, the general framework is able to capture arbitrary kinds of social dilemmas (Supplementary Figs. 9 and 10).

The players' decisions in each round depend on the previous history of play and on the players' strategies. To quantify the effects of strategic spillovers between different games, we distinguish two versions of multichannel games. The unlinked case (Fig. 1b) serves as a control scenario. Here, any spillovers are excluded. Each player's action in game $k$ may only depend on the previous history of game $k$. In contrast, in the linked case (Fig. 1c), a player's action in game $k$ may depend on the outcome of other games as well.

To make a computational analysis feasible, we suppose players are restricted to strategies of some given complexity. Throughout most of the main text, we assume players use reactive strategies. That is, their actions in any given round may depend on their co-player's actions in the previous round, but they are independent of all other aspects. In the unlinked case, we define reactive strategies as the elements of the set

$$\mathcal{R}_U = \left\{ \mathbf{p} = (p_{a_1}^1; p_{a_2}^2; \dots ; p_{a_m}^m)_{a_k \in \{C,D\}, k \in \{1,\dots,m\}} \ \middle| \ p_{a_k}^k \in [0,1] \text{ for all } k \right\}. \quad (6)$$

Here, $p_{a_k}^k$ is the player's cooperation probability in game $k$, which depends on which action $a_k \in \{C, D\}$ the co-player has chosen in the previous round of that game. For $m = 2$, the elements of $\mathcal{R}_U$ take the form of the four-dimensional vector represented in Eq. (1). In the linked case, reactive strategies are the elements of the set

$$\mathcal{R}_L = \left\{ \mathbf{p} = (p_{\mathbf{a}}^k)_{\mathbf{a} \in \{C,D\}^m, k \in \{1,\dots,m\}} \ \middle| \ p_{\mathbf{a}}^k \in [0,1] \text{ for all } k \text{ and } \mathbf{a} \right\}. \quad (7)$$

Here, $p_{\mathbf{a}}^k$ is again the player's conditional cooperation probability in game $k$. However, this time, this probability depends on the co-player's last actions in all $m$ games, represented by the vector $\mathbf{a} = (a_1, \dots, a_m) \in \{C, D\}^m$. For $m = 2$, reactive strategies take the form of eight-dimensional vectors, as represented in Eq. (2). For the simulations, we assume that players can choose any strategy in either $\mathcal{R}_U$ (in the unlinked case) or $\mathcal{R}_L$ (in the linked case).

In addition to reactive strategies, we have also run simulations in which players can choose among all memory-1 strategies (Fig. 4 and Supplementary Fig. 7). Here the players' actions depend on their co-player's previous decisions and on their own previous decisions. We formally define the respective strategy spaces for the unlinked and the linked case in Supplementary Note 4. As with reactive strategies, simulations suggest that when players are able to link their games, they achieve more cooperation in both games (Fig. 4 and Supplementary Fig. 7).

We consider infinitely many rounds in the limit of no discounting. For each game $k$, we define the associated repeated-game payoff as the limit of the player's average payoff per round (for the cases we consider, the existence of this limit is guaranteed). A player's payoff in the multichannel game is defined as the sum over all her $m$ repeated-game payoffs.

We may sometimes assume that a player misimplements her intended action. Specifically, with probability $\varepsilon$, a player who intends to cooperate instead defects, and conversely a player who intends to defect cooperates. In addition to making the model more realistic, implementation errors ensure that payoffs are well-defined, independent of the outcome of the very first round of the game[7,23]. Our simulation results are robust with respect to the exact magnitude of this error rate, provided that errors are sufficiently rare for the player's strategies to have an impact (Supplementary Fig. 6d). For further details, see Supplementary Note 2.

**Evolutionary dynamics.** To model the evolution of strategies over time, we consider a pairwise comparison process[41,42] in a population of size $N$. Each player interacts with every other population member in the respective multichannel game. A player's payoff in the population game is defined as her average payoff across all multichannel games she participates in.

To consider the most stringent case for the evolution of cooperation, initially each player adopts the strategy ALLD. That is, for any outcome of the previous round, each player's conditional cooperation probability is zero. Then, in each time step of the simulation, one population member is chosen at random to update her strategy. There are two different updating methods. With probability $\mu$ (referred to as mutation rate), the chosen player engages in random strategy exploration. In that case, the player randomly picks a new strategy from the set of all available strategies (for reactive strategies, this set is $\mathcal{R}_U$ in the unlinked case, and it is $\mathcal{R}_L$ in

the linked case; for memory-1 strategies the respective sets are defined analogously).

Alternatively, with probability $1 - \mu$, the chosen player picks a random role model from the population. If the focal player's payoff is $\pi_F$ and the role model's payoff is $\pi_R$, the focal player adopts the role model's strategy with probability[54]

$$\rho = \frac{1}{1 + \exp[-s(\pi_R - \pi_F)]}. \quad (8)$$

The parameter $s \geq 0$ is called the strength of selection[55]. It reflects to which extent the focal player aims to achieve higher payoffs when updating her strategy. If $s = 0$, payoffs are irrelevant and imitation occurs at random. In the other limit when $s \to \infty$, a player always updates when considering a role model with higher payoff.

Over time, the interaction of random strategy exploration and imitation yields an ergodic process on the space of all possible population compositions. For our simulations, we implement this process in the limit of rare mutations, $\mu \to 0$, which allows for an easier computation of the dynamics[43–46]. The respective code is provided in Supplementary Note 5. As illustrated in Supplementary Fig. 6c, we obtain similar results for larger mutation rates, provided mutations are not too common compared to imitation events.

**Analytical results for reactive strategies.** To complement our numerical simulations, we have mathematically characterized three different classes of Nash equilibria when each game $k$ is a donation game. A strategy $\mathbf{p}$ is a Nash equilibrium if no player has an incentive to deviate if every other player adopts $\mathbf{p}$. We note that deviations need to be interpreted broadly: for a strategy to be a Nash equilibrium, no other strategy is allowed to yield a higher payoff, not even a strategy of higher complexity as strategy $\mathbf{p}$. We call a strategy self-cooperative in game $k$ if its cooperation rate against itself in game $k$ approaches one in the limit of no errors. Similarly, the strategy is self-defective in game $k$, if the respective cooperation rate approaches zero. Based on these notions, we define partners, semi-partners, and defectors as follows. A strategy is a partner if it is a Nash equilibrium and if it is self-cooperative in all games $k$. Similarly, a strategy is a defector if it is a Nash equilibrium and if it is self-defective in every game. Finally, the strategy is a game $k$ semi-partner, if it is a Nash equilibrium and if it is self-cooperative in game $k$ but self-defective in all other games.

Within the space of reactive strategies, we can characterize the partners, semi-partners, and defectors in the linked case as follows. To simplify notation, we introduce an indicator variable $e_{\mathbf{a}}^k$. Its value is one if the $k$-th entry of the co-player's action profile $\mathbf{a} = (a_1, \dots, a_m)$ is C and it is zero otherwise. Using this notation, we obtain (for details, see Supplementary Note 3, Propositions 1–3):

1. A strategy $\mathbf{p} \in \mathcal{R}_L$ that is self-cooperative in each game $k$ is a partner if and only if $\sum_{k=1}^m b_k \cdot (1 - p_{\mathbf{a}}^k) \geq \sum_{k=1}^m c_k \cdot (1 - e_{\mathbf{a}}^k)$ for all co-player's action profiles $\mathbf{a} \in \{C, D\}^m$.
2. A strategy $\mathbf{p} \in \mathcal{R}_L$ that is self-defective in each game $k$ is a defector if and only if $\sum_{k=1}^m b_k \cdot p_{\mathbf{a}}^k \leq \sum_{k=1}^m c_k \cdot e_{\mathbf{a}}^k$ for all co-player's action profiles $\mathbf{a} \in \{C, D\}^m$.
3. A strategy $\mathbf{p} \in \mathcal{R}_L$ that is self-cooperative in game $k$ but self-defective in all other games is a game $k$ semi-partner if and only if $b_k \cdot (1 - p_{\mathbf{a}}^k) - c_k \cdot (1 - e_{\mathbf{a}}^k) \geq \sum_{l \neq k} b_l \, p_{\mathbf{a}}^l - \sum_{l \neq k} c_l \, e_{\mathbf{a}}^l$ for all co-player's action profiles $\mathbf{a} \in \{C, D\}^m$.

In the case of $m = 2$, the condition for partners simplifies to condition (5) in the main text. The above results are also illustrated in Supplementary Fig. 2.

Similarly, we can characterize partners, semi-partners, and defector among the reactive strategies for the unlinked case (for details, see Supplementary Note 3, Proposition 4).

1. A strategy $\mathbf{p} \in \mathcal{R}_U$ that is self-cooperative in each game $k$ is a partner if and only if $p_D^k \leq 1 - c_k/b_k$ for all games $k$.
2. A strategy $\mathbf{p} \in \mathcal{R}_U$ that is self-defective in each game $k$ is a defector if and only if $p_C^k \leq c_k/b_k$ for all games $k$.
3. A strategy $\mathbf{p} \in \mathcal{R}_U$ that is self-cooperative in game $k$ and self-defective in all other games is a game $k$ semi-partner if and only if $p_D^k \leq 1 - c_k/b_k$ and $p_C^l \leq c_l/b_l$ for all $l \neq k$.

For the special case of $m = 2$ games, the respective condition for partners yields condition (4) in the main text. Supplementary Fig. 1 provides a graphical illustration. As one may expect, when there is only $m = 1$ game, the respective conditions in the linked case coincide with the respective conditions for the unlinked case. In particular, the condition for partner strategies yields a maximum cooperation rate after defection of $p_D^k = 1 - c_k/b_k$, which recovers the value of the classical Generous Tit-for-Tat strategy[3,4]. We can also use the above conditions for partners, semi-partners, and defectors to calculate how abundant the respective strategies are among all reactive strategies. This calculation confirms that for most parameter values, partners are more abundant when games are linked (see Supplementary Fig. 3 and Supplementary Note 3 for details).

**Analytical results for memory-1 strategies.** The simulations for memory-1 players in Fig. 4 suggest that in the unlinked case, players establish little cooperation when $b_k < 2c_k$. In contrast, in games with $b_k > 2c_k$, cooperation seems to be maintained with the strategy Win-Stay Lose-Shift (WSLS). A player with that

strategy cooperates if and only if either both players have cooperated in the previous round of the respective game, or if no one did. In the linked case, evolving strategies resemble a different strategy, which we term CIC. Players with this strategy use in each round the same action in all games they participate in. This action is cooperation if and only if in each game, players used the same action in the last round; otherwise they defect.

We can characterize for which parameter values $b_k$ and $c_k$ these two strategies are subgame perfect equilibria. A subgame perfect equilibrium is a refinement of the Nash equilibrium: players are required not to have an incentive to deviate after any previous history of play[56]. We obtain the following conditions (Supplementary Note 4, Proposition 5).

1. WSLS is a subgame perfect equilibrium if and only if $b_k \geq 2c_k$ for all $k$.
2. CIC is a subgame perfect equilibrium if and only if $\sum_k b_k \geq 2\sum_k c_k$.

The two conditions again reflect one reason why full cooperation is easier to sustain in the linked case. Unlinked strategies like WSLS require that the benefit satisfies $b_k \geq 2c_k$ in every single game. In contrast, in the linked case, CIC only requires that this condition is met on average, across all games. In particular, players may use cooperation in high-benefit games (with $b_k > 2c_k$) as a means to achieve cooperation in low-benefit games (with $b_k < 2c_k$).

**Reporting summary**. Further information on research design is available in the Nature Research Reporting Summary linked to this article.

## Data availability
The raw data generated with these computer simulations is available from the authors upon reasonable request.

## Code availability
All simulations and numerical calculations have been performed with MATLAB R2014A. We provide the respective code in Supplementary Note 5.

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

## Acknowledgements

M.A.N. was supported by the Army Research Laboratory (grant W911NF-18-2-0265), the Bill & Melinda Gates Foundation (grant OPP1148627), the John Templeton Foundation (grant 61443), and the Office of Naval Research (grant N00014-16-1-2914). C.H. acknowledges generous support by the Max Planck Society. Open access funding provided by Projekt DEAL.

## Author contributions

K.D., O.H., M.A.N., and C.H.: designed the research; K.D., O.H., M.A.N., and C.H.: Performed the research; K.D., O.H., M.A.N., and C.H.: wrote the paper.

## Competing interests

The authors declare no competing interests.
