## [Peer Review File · Nature Communications]

Reviewers' Comments:

Reviewer #1:

Remarks to the Author:

This paper studies the evolution of cooperation in multichannel games. The authors introduce a general framework of multichannel games, where two or more individuals interact with each other simultaneously over different channels, and each channel is a repeated game. In such multichannel games, a choice of strategies in one channel can affect strategic decisions in another. The authors show through analytical calculations and evolutionary simulations that this strategic linking enhances cooperative behavior in a variety of social dilemmas, including the donation game. The approach introduced in this paper is a natural generalization of previous approaches to studying the evolution of cooperation in repeated games and it represents a type of situation that can be expected to occur frequently in the real world. The paper is well-written and will be of interest to a wide range of researchers working on game theory and the evolution of cooperation. In addition, the Supplementary Information is comprehensive and informative.

I have only a few minor comments that may help to improve the paper.

1. Since many of the simulations are for large ensembles it would be interesting to have some sense of the statistical distribution of the results. Therefore, it would probably be better to replace the bar graphs in figures 2, 4 and S9 with box-and-whisker plots.
2. The authors have considered in addition to the donation game also the snowdrift game and the volunteer's dilemma, however, it might also be interesting to consider the stag hunt game. While the stag hunt game is something of a Cinderella in game theory, it might nevertheless be very interesting to study this game in the context of multichannel games. It would be natural to consider both scenarios in which players engage in two stag hunt games and in a stag hunt game combined with a donation game. The sculling game introduced in (Iyer, S. and Killingback, T., 2016, Evolution of cooperation in social dilemmas on complex networks. PLoS Computational Biology 12 (2), e1004779) represents a version of the stag hunt game that is analogous to the donation game and snowdrift game, and so may be well suited to such a study.
3. On a technical point, the authors calculate the payoffs in the infinitely iterated games from the stationary distribution of a Markov chain - do they have a formula for the payoffs that is analogous to the Press-Dyson determinant formula?

Reviewer #2:

Remarks to the Author:

In this well-written paper the authors have extended the evolutionary games by allowing the players to use different (including correlated) reactive strategies in games played simultaneously. This feature eliminates the similarity to spin systems in physical models. Using numerical simulations they consider multi-agent models where the pair interactions are restricted to a set of repeated donation games. It is found that if the agents can link their their concurrently ongoing interactions then they learn to coordinate their behavior that will be beneficial for them. These processes are discussed for several important cases. I think that the present models include relevant novelties that are present in real systems and this idea will initiate a large scale of further activities. Thus I recommend the publication of this interesting and important manuscript after some minor corrections detailed below:

In the present analyses the pair interactions are limited to donation games that do not include

coordination (or anti-coordination) type interactions which can introduce additional Nash equilibria and modify the evolutionary process. I suggest specifying clearly the interactions by substituting "donation game" for "game" in the Abstract, Introduction, and Discussion.

We would like to thank the editor and the two referees. The feedback has been very useful and constructive. Of course, we are also happy to learn that the referees generally find our manuscript to be suitable for publication in Nature Communications.

In the meanwhile, we have incorporated all their remaining suggestions. Please find our detailed point-by-point reply below.

Reviewer #1:

This paper studies the evolution of cooperation in multichannel games. The authors introduce a general framework of multichannel games, where two or more individuals interact with each other simultaneously over different channels, and each channel is a repeated game. In such multichannel games, a choice of strategies in one channel can affect strategic decisions in another. The authors show through analytical calculations and evolutionary simulations that this strategic linking enhances cooperative behavior in a variety of social dilemmas, including the donation game. The approach introduced in this paper is a natural generalization of previous approaches to studying the evolution of cooperation in repeated games and it represents a type of situation that can be expected to occur frequently in the real world. The paper is well-written and will be of interest to a wide range of researchers working on game theory and the evolution of cooperation. In addition, the Supplementary Information is comprehensive and informative.

Reply: Thank you very much. We appreciate this positive feedback.

I have only a few minor comments that may help to improve the paper.

1. Since many of the simulations are for large ensembles it would be interesting to have some sense of the statistical distribution of the results. Therefore, it would probably be better to replace the bar graphs in figures 2, 4 and S9 with box-and-whisker plots.

Reply: We fully agree. In some of our figures, including the ones mentioned above, we show averages over many simulations. Between different simulation runs, there can be quite some variation. Instead of a simple bar plot, readers might thus get a clearer picture if we use a representation that also visualizes the distribution of the underlying simulation data.

*However, for our type of data, box-and-whisker plots are not ideal either. The problem is easiest to explain for **Figure 4**. In that figure, we visualize how often players cooperate on average after each possible outcome of the previous round. For the simulations, we assumed that players can choose among all deterministic memory-1 strategies. The cooperation probabilities of such strategies can only assume two possible values, zero or one. For such data, the interquartile range would either degenerate to a single point (either zero or one), or it would cover the entire unit interval (from zero to one).*

The output variables in **Figures 2** and **S9** are not binary; the depicted probabilities and abundances can assume any value between 0 and 1. But also here, box-and-whisker plots would result in a similar problem. In many cases, the respective abundances either cluster in the vicinity of zero, or in the vicinity of one. As a consequence, the interquartile range would again be very small or very large. Maybe more importantly, already small changes in the data could change the appearance of the resulting box-and-whisker plot entirely.

Changes: Instead of box-and-whisker plots, we thus opted for an alternative representation of our data. As before, we use bars to depict mean values. In addition, we use dots to depict the distribution of 100 representative simulation outcomes in **Figures S2** and **S9**. These dots provide readers with an intuitive illustration how the underlying distribution looks like.

Because the individual data points in **Figure 4** can only assume two possible values (either zero or one), the bar diagram in this figure already captures the underlying distribution completely. We therefore do not display any individual dots here.

2. The authors have considered in addition to the donation game also the snowdrift game and the volunteer's dilemma, however, it might also be interesting to consider the stag hunt game. While the stag hunt game is something of a Cinderella in game theory, it might nevertheless be very interesting to study this game in the context of multichannel games. It would be natural to consider both scenarios in which players engage in two stag hunt games and in a stag hunt game combined with a donation game. The sculling game introduced in (Iyer, S. and Killingback, T., 2016, Evolution of cooperation in social dilemmas on complex networks. PLoS Computational Biology 12 (2), e1004779) represents a version of the stag hunt game that is analogous to the donation game and snowdrift game, and so may be well suited to such a study.

Reply: This is a very helpful suggestion. Indeed, throughout our manuscript we have focused on cooperation in strict social dilemmas. In all games considered, players had some incentive to deviate from mutual cooperation. However, interesting tensions between cooperation and defection can arise even in games in which mutual cooperation is stable. For example, when both behaviors form an equilibrium of the game, players may have problems to coordinate on the most beneficial equilibrium. We fully agree with the reviewer that this is an interesting problem to consider.

Changes: In response to this suggestion, we have extended our **Figures S9** and **S10**. In **Figure S9** we explore the dynamics when players engage in two coordination games. As suggested by the reviewer, each game takes the form of a different sculling game. In addition, in **Figure S10** we explore the dynamics when one game takes the form of a sculling game, whereas the other one is a prisoner's dilemma. Our simulations suggest that in both cases, multichannel games are now somewhat less effective. People tend to be highly cooperative already in the unlinked sculling games. Allowing for linkage can sometimes lead to a small reduction in overall cooperation rates. In our revised manuscript, we address these additional results for coordination games in the discussion section. In addition, we explain them in more detail in **SI Section 4.2**.

3. On a technical point, the authors calculate the payoffs in the infinitely iterated games from the stationary distribution of a Markov chain - do they have a formula for the payoffs that is analogous to the Press-Dyson determinant formula?

*Reply: This is a very interesting question. Indeed, to compute the players' payoffs, we have calculated the stationary distribution of the respective Markov chain (as described in **SI Section 2.3**). In a quite influential paper, Press and Dyson (PNAS 2012) suggested an alternative formula to calculate payoffs for the special case of an iterated prisoner's dilemma. Their approach is based on determinants of certain matrices.*

The formalism of Press and Dyson is actually more general than most people seem to appreciate. Although they formally only discuss the prisoner's dilemma, the main idea of their determinant formula applies to all games that can be described by a Markov chain. In particular, one can use their approach to derive a payoff formula for multichannel games.

*Changes: In **SI Section 2.3**, we now derive an explicit payoff formula with determinants, similar to the one used by Press and Dyson. Please note, however, that computing these determinants is of the same computational complexity as calculating the stationary distribution of the respective Markov chain. For this reason, all actual payoff computations are still performed with our original algorithm.*

Reviewer #2:

In this well-written paper the authors have extended the evolutionary games by allowing the players to use different (including correlated) reactive strategies in games played simultaneously. This feature eliminates the similarity to spin systems in physical models. Using numerical simulations they consider multi-agent models where the pair interactions are restricted to a set of repeated donation games. It is found that if the agents can link their their concurrently ongoing interactions then they learn to coordinate their behavior that will be beneficial for them. These processes are discussed for several important cases. I think that the present models include relevant novelties that are present in real systems and this idea will initiate a large scale of further activities. Thus I recommend the publication of this interesting and important manuscript after some minor corrections detailed below:

Reply: Thank you for this encouraging feedback!

In the present analyses the pair interactions are limited to donation games that do not include coordination (or anti-coordination) type interactions which can introduce additional Nash equilibria and modify the evolutionary process. I suggest specifying clearly the interactions by substituting "donation game" for "game" in the Abstract, Introduction, and Discussion.

Reply: We agree that while our framework is general, most of our analytical results are derived for the special case of donation games. However, we do explore other types of games with simulations, as depicted in **Figures S9** and **S10**. In particular, these figures include an example of a coordination game (the sculling game suggested by reviewer #1) and two examples of anti-coordination games (the snowdrift game and the volunteer's dilemma).

Changes: In light of these numerical results for other types of games, we prefer to speak of games more generally. However, we now state more clearly – in the abstract, the introduction, and the discussion – that our analytical results are derived for donation games.

Reviewers' Comments:

Reviewer #1:

Remarks to the Author:

The authors have carefully and thoughtfully revised this paper and it is now suitable for publication.